

# Evaluation and enhancement of a low-cost NDIR $CO_2$ sensor

Cory R. Martin[1], Ning Zeng[1,2], Anna Karion[3], Russell R. Dickerson[1,2], Xinrong Ren[1,4], Bari N. Turpie[1], Kristy J. Weber[1,5]

[1]Department of Atmospheric and Oceanic Science, University of Maryland, College Park, MD 20742, USA
[2]Earth System Science Interdisciplinary Center, University of Maryland, College Park, MD 20742, USA
[3]National Institute of Standards and Technology, Gaithersburg, MD 20899, USA
[4]Air Resources Laboratory, National Oceanic and Atmospheric Administration, College Park, MD 20740, USA
[5]Now at: Department of Geography, University of Colorado at Boulder, Boulder, CO 80309, USA

*Correspondence to*: Ning Zeng (zeng@umd.edu)

**Abstract.** Non-dispersive infrared (NDIR) sensors are a low-cost way to observe carbon dioxide concentrations in air, but their specified accuracy and precision are not sufficient for some scientific applications. An initial evaluation of six SenseAir K30 carbon dioxide NDIR sensors in a lab setting showed that without any calibration or correction, the sensors have an individual root mean square (RMS)
error between ~5 to 21 parts per million (ppm) compared to a research-grade greenhouse gas analyzer using cavity enhanced laser absorption spectroscopy. Through further evaluation, after correcting for environmental variables with coefficients determined through a multivariate linear regression analysis, the calculated difference between the each of six individual K30 NDIR sensors and the higher-precision instrument had for one minute data a standard deviation of between 1.6 ppm and 4.4 ppm. The median
standard deviation improved from 8.08 for off the shelf sensors to 1.89 ppm after correction and calibration, demonstrating the potential to provide useful information for ambient air monitoring.

## 1 Introduction

Carbon dioxide ($CO_2$) is a major greenhouse gas, with fundamental importance to Earth's climate. Since measurements started at the Mauna Loa Observatory in the 1950s (Keeling et al., 2005), the global mean
concentration of $CO_2$ has steadily risen from the preindustrial dry air mole fraction of approximately 280 $\mu$mol mol$^{-1}$ of dry air (parts per million, or ppm) to today's level exceeding 400 ppm. These observations, both from flask samples and state-of-the-art continuous measurement instruments, have a typical compatibility of ~0.1 ppm, recommended for observations at background global network sites (World Meteorological Organization, 2013). Flask-based measurements require observers to collect samples
subsequently transported to a lab for analysis, at significant cost. Continuous in-situ $CO_2$ analyzers located at towers do not suffer from these regular costs, but these high-precision analyzers can cost upwards of $100,000 per site, plus any additional installation costs including adding inlets to towers or rooftops. High-



accuracy carbon dioxide observations are thus relatively sparse compared to other climatological variables such as temperature and precipitation. Recent research efforts have focused more locally and on the use of networks of observing sites that use instrumented towers similar to what is used for global monitoring, but applied to the urban environment (Pataki et al., 2003; Briber et al., 2013; Kort et al., 2013; McKain et al.,

2012; Turnbull et al., 2015). High-accuracy observations from these tower sites are then used to create inversions to estimate the total greenhouse gas flux from the urban area in question (McKain et al., 2012; Bréon et al., 2015; Lauvaux et al., 2016). However, due to the cost of these networks being comparable to ones at the global scale, the observation towers are still sited at a relatively low density of typically 8 to 12 sites in a single metropolitan area (Turnbull et al., 2015; Kort et al., 2013). A higher spatial density of

observations in these urban regions has been shown to better constrain the inversion estimates, even if the absolute uncertainty of the observations is higher (Wu et al., 2016; Lopez-Coto et al., under review), but a trade-off between total network cost and inversion constraint must be balanced.

Recently, however, a wave of small, low-cost sensors, some of which measure trace gases or particulate

matter, in addition to traditional meteorological variables, using various technologies have become commercially available. Evaluation and implementation of these sensors is quite promising (Eugster and Kling, 2012; Holstius et al., 2014; Piedrahita et al., 2014; Young et al, 2014; Wang et al., 2015; Shusterman et al., 2016). Many of these instruments are based on electrochemical reactions to measure the concentrations of trace gases. With the advent of widely available and low cost mid-IR light sources and

detectors, a small group of non-dispersive infrared (NDIR) $CO_2$ sensors have also become commercially available. They are designed for use in a number of applications including ventilation control, agricultural and industrial applications, and inclusion in stand-alone commercial products. Additionally, with the high volume of possible applications, these small NDIR $CO_2$ sensors are affordably priced on the order of $100 to $200 per sensor. Previous studies have compared some of these NDIR $CO_2$ devices and concluded that

after application of some type of calibration procedure, some of these devices can provide reasonably accurate measurements of ambient $CO_2$ concentrations (Hurst et al., 2011; Yasuda et al., 2012; Shusterman et al., 2016).

In this paper, one of these small NDIR $CO_2$ devices is assessed by determining its precision with and

without environmental corrections. Section 2 describes the $CO_2$ sensor and its Allan variance, the other instruments included in the system, and the data collection and processing methodology. Section 3 describes the calibration and shows the stability of the reference high-precision gas analyzer, and the initial results from the NDIR sensor are shown in Sect. 4. In Section 5, two methods are described to determine



functional relationships and coefficient values to correct the observed values of the instrument for environmental variables and Sect. 6 discusses the potential utility of observations from this sensor after correction and temporal averaging.

## 2 Instruments and methods

To test the validity of using low-cost sensors for scientific applications, a sensor package was implemented consisting of various off-the-shelf components. The K30 sensor module (K30), manufactured by the Swedish company, SenseAir, is the low-cost NDIR $CO_2$ observing instrument[1]. The K30 is a microprocessor-controlled device with on-board signal averaging, has a measurement range of 0 to 10,000 ppm, observation frequency of 0.5 Hz, and resolution of 1 ppm. The manufacturer's stated accuracy of the

K30 sensor is given as ±30 ppm ±3 % of reading (SenseAir, 2007). Additional NDIR sensors were initially evaluated before selecting the K30, including the COZIR ambient sensor and Telaire T6615, having accuracies described as being ±50 ppm ±3 % and ±75 ppm respectively (Gas Sensing Solutions, 2014; General Electric, 2011). The K30 was chosen not only because of its highest manufacturer's specified accuracy, but also because of observed reliability and consistency with observations. In addition to $CO_2$,

temperature, relative humidity, and pressure readings are recorded using a breakout board purchased from Adafruit. This board features a Bosch Sensortec BME280, which according to the manufacturer's datasheet has an average absolute accuracy of ±1 ℃, ±3 %, and ±1 hPa, and an output resolution of 0.1 ℃, 0.008 % and 0.01 hPa, respectively (Bosch Sensortec, 2015).

To compare the performance of the K30 to better-performing research instrumentation, a greenhouse gas analyzer based on cavity enhanced absorption spectrometry (CEAS) was used as the control. The LGR-24A-FGGA fast greenhouse gas analyzer (FGGA) from Los Gatos Research (LGR, San Jose, CA) provides $CO_2$, $CH_4$, as well as water vapor mixing ratios at a frequency of 0.5 Hz and has an un-calibrated uncertainty of less than one percent (Los Gatos Research, 2013). The FGGA was connected to a tee

connection, to allow either ambient air or a calibration source (during calibrations) to be sampled continuously by the analyzer at a flow rate of 400 standard mL min$^{-1}$. Calibrations for $CH_4$ and $CO_2$ were conducted using several NIST-

---

[1]Certain commercial equipment, instruments, or materials are identified in this paper in order to specify the experimental procedure adequately. Such identification is not intended to imply recommendation or endorsement by the National Institute of Standards and Technology, nor is it intended to imply that the materials or equipment identified are necessarily the best available for the purpose.



certified standard mixtures every 23 to 47 hours for a period of one month with molar mixing ratios ranging from 1869.6 parts per billion (ppb) to 2159.4 ppb for $CH_4$ and from 369.19 ppm to 429.68 ppm for $CO_2$. See Sect. 3 for details and results of this calibration period.

It is important to note that there are differences in how CEAS works compared to NDIR, most notably that the FGGA and other CEAS instruments have a controlled cavity where pressure and temperature are kept nearly constant, removing potential environmental interference and the need for corrections, whereas the NDIR K30 works in the ambient environment without any mechanism for keeping temperature or pressure constant. Additionally, the FGGA implements a water vapor correction on its greenhouse gas

concentrations to estimate the dry gas mixing ratio, while the K30 makes no water vapor corrections. To increase the effective path length, both the K30 and FGGA use mirrors, but the FGGA system uses highly reflective mirrors that allow for an effective path length that is many times longer than that of the K30. Additionally, the CEAS instrument determines the concentration of a gas by how long it takes for the signal to degrade inside the cavity (the e-folding time), whereas an NDIR sensor merely measures the intensity of

the signal received relative to the total intensity emitted.

For data collection, a Raspberry Pi (RPi) computer is used (Raspberry Pi Foundation, 2015). The RPi is a credit card sized (approximately 6 x 9 cm) computer running a full Linux distribution, allowing for easy customization and usability, that is priced at around $25. The K30 is connected to the RPi over Universal

Asynchronous Receiver/Transmitter (UART) Serial, and the BME280 over Inter-Integrated Circuit ($I^2C$) serial. An image of the complete sensor package is available in Fig. 1. Data is archived on the RPi and uploaded to a centralized data storage and processing server. The FGGA collects and archives its own data, but an RPi is used here as well to collect the data from the FGGA over a local area network and transfer it to the same centralized server. The added computational power of a Raspberry Pi over traditional data

loggers allows for the ability to archive two levels of data: the raw data collected every two seconds, and one-minute averages.

Archiving and comparing multiple datasets proved to be challenging, so steps are taken to ensure that each compared value is at the same observed time. Because of various complications, the data collection times

of each K30 sensor package and the FGGA are asynchronous. Additionally, power issues can corrupt parts of the plain text data files stored on the RPi's SD card with random characters. Thus, a post-processing procedure has been developed that filters extraneous characters, and then each dataset is synchronized



based on recorded time stamps and averaged over selected time periods. These new datasets can then be directly compared without missing or out of phase data points.

## 2.1 K30 Allan variance

Allan variance (Allan, 1966) is a measure of the time-averaged stability between consecutive measurements or observations, often applied to clocks and oscillators. In addition, an Allan variance analysis can be used to determine the optimum averaging interval for a dataset to minimize noise without sacrificing signal. Figure 2 shows the Allan deviation (the square root of the variance) for one K30's raw two-second data when exposed to a known reference gas. The original two-second data shows the

maximum noise, with a standard deviation comparable to the manufacturer's specifications of ±30 ppm, but averaging for even ten seconds drops the variance significantly. According to this analysis, the optimum averaging time, when the Allan variance is at a minimum (Langridge et al., 2008), is approximately three minutes; longer averaging times to now reduce the noise. For the subsequent analysis, an averaging time of one minute is used, as the Allan variance is only slightly higher than for three minutes, and one minute

observations allow for resolution of atmospheric variability at shorter time scales.

## 2.2 Experiment

The need to quickly and effectively evaluate a relatively large number of sensors under conditions with relatively stable $CO_2$ led to the use of a rooftop observation room on the University of Maryland campus in College Park, MD. Because this rooftop room had limited access, and it was not part of the building's

HVAC system, it served as an ambient evaluation chamber with minimal influence from human respiration. The room was slightly ventilated for the entire evaluation period to allow outside air to slowly diffuse into the room, with a small household box fan also in the room to ensure that the air was well mixed. The room also features a small, independent heating and cooling unit, but it was only used to keep the room from exceeding a certain temperature, thus the room was not fully temperature controlled. Even with this control,

the diurnal fluctuations of temperature in the room were similar to that of the outdoor environment. This ventilation strategy was intentional so that the room then mimicked the ambient $CO_2$ concentration of the surrounding atmosphere, and approximated the outdoor temperature and humidity, while protecting instruments from direct sunlight, extreme temperatures, and inclement weather. This provided an advantage over controlled tests in a laboratory setting in that rather than just a multi-point calibration, comparing

datasets over ambient concentrations and environmental conditions allowed for a realistic evaluation of these instruments in more real world scenarios.





For a continuous period of approximately four weeks in spring 2016, six K30 sensor packages as described in Sect. 2 were deployed alongside the LGR FGGA in the rooftop room, all sampling room air. The FGGA was also connected to a mass flow controller and standard tank to periodically provide a reference for stability (details in Sect. 3). For the reference dataset, the dry carbon dioxide ($CO_{2\ dry}$) output calculated by

the FGGA was used. This output includes an applied correction to the mole fraction of $CO_2$ to give the dry air mole fraction in ppm. The raw $CO_2$ values were recorded from each K30, temperature and pressure were recorded from each BME280 sensor, and water vapor mole fraction was also recorded by the FGGA. All of the observations were recorded every two seconds, and averaged into one minute values. The next two sections describe the stability of the FGGA as well as the initial comparison between the K30 and FGGA

observations.

## 3 Los Gatos evaluation and correction

To evaluate the K30 NDIR sensor performance compared to a research-grade analyzer, first the control dataset needs to be calibrated and corrected for drift. To calibrate the FGGA, after the experiment concluded the dataset was corrected using a two-point calibration curve derived from using two NIST-

traceable gas standards, one with a $CO_2$ mole fraction of 369.19 ppm, and the other with a mole fraction of 429.68 ppm. A linear fit was then assumed between the two calibration points, with the recorded values as the dependent variable and the NIST-assigned tank values as the independent variable. Once the coefficients were determined, the entire FGGA dataset was then corrected for further analysis.

Additionally, there was a need to quantify any drift as well as to correct the FGGA reported values based on known gas concentrations. During the experiment period, the FGGA was attached to a tee connector, which pulled ambient air from the aforementioned evaluation chamber using its included pump most of the time, but would also receive periodic calibration every 23 to 47 hours for a period of either 10 minutes or one hour using a reference tank of breathing air connected to a Dasibi Model 5008 calibrator. This

breathing air tank is assumed to have a fixed $CO_2$ mole fraction, which was estimated by using the FGGA to be 463.7 ppm and was used to quantify and subtract the drift of the FGGA over the comparison period.

In Fig. 3, the ambient data from the FGGA has been filtered out to show only each calibration period performed during the month long experiment. The data during each calibration period was averaged (either

a total of 10 minutes or one hour depending on the calibration period) and the averages are plotted on Fig. 3. While there is some small variation in the mean mole fraction observed during each calibration from day-to-day, there was an upward trend in the recorded value, by over 1.2 ppm over a 30-day period. This





observed drift, while not insignificant, is well within the manufacturer's specifications for this analyzer. However, the observed standard deviation of the two-second points used in each average (the error bars on Fig. 3) remained relatively constant throughout the period at around ±0.4 ppm. This high-frequency noise is not a problem for the analysis with the K30 sensor because both datasets are averaged to one minute values,

which removes most, if not all, of this noise. For comparisons between the K30s in the remainder of this paper, the FGGA drift is corrected by first linearly interpolating between each calibration point in time and then subtracting from the FGGA dataset the difference from the tank's assigned value of 463.7 ppm.

## 4 Initial K30 results

Figure 4 shows the original time series of data recorded during the evaluation experiment described in Sect.
2.2. The top panel shows raw $CO_2$ mole fractions reported by six K30 sensors as well as the LGR FGGA analyzer, each of which is located in the same rooftop evaluation chamber. The middle panels show the reported atmospheric pressure and temperature values from one BME280 sensor, and the water vapor mole fraction from the FGGA. Then, the bottom panel is the difference between the original recorded K30 value and the corrected FGGA recorded $CO_2$ mole fraction with the calibration periods removed.

Over this four-week period, the FGGA observed an ambient variation of $CO_2$ with an average value of just over 423 ppm, and a standard deviation of just under 21 ppm. There is distinct synoptic variation in the diurnal cycle observed, with the magnitude varying from as little as 10 ppm over 24 hours to more than 100 ppm. Each of the K30s was successfully able to resolve the ambient variations in $CO_2$ over this evaluation
period, although none of the K30s matched the FGGA perfectly in both absolute concentration and relative change. However, without any correction or calibration, each K30 was well within the manufacturer's stated uncertainty of ±30 ppm ±3% of the reading.

From the difference plot (Fig. 4, bottom panel), there are some important things to note. First and foremost,
each individual K30 sensor has a distinct zero offset. A few of the sensors are approximately the same as the FGGA, but many can have an offset that is as much as 5 % (20 ppm) from the LGR FGGA. The differences between each K30 and the FGGA all have standard deviations between 4 ppm to 6 ppm and root mean square errors (RMSE) between 5 ppm to 21 ppm. This means that after accounting for the offset of each individual K30, the practical accuracy of the K30 $CO_2$ sensor can be within 1 % of the observed
concentration. Secondly, each K30 difference time series appears to feature two wave patterns, one with a period of around one week, and another with a period of approximately one day. Given that the cycles seem fairly consistent and are present in each K30, this suggests that the difference between the recorded values



from the FGGA and each K30 is not random, but instead that there are external factors that can be assessed for potential compensation in the K30 response.

## 5 Environmental correction

In Fig. 4, the difference between the FGGA and each K30 is shown in the bottom panel below time series of environmental data from the evaluation chamber. Just like in the difference plot, each of the environmental variables features two distinct time scales of variability. There is a diurnal cycle of each variable, as well as synoptic-scale variability attributed to weather systems that occurs on the order of one week. Because the observed $CO_2$ differences and the environmental variables are correlated on both short and long time scales, statistical regression methods were used to correct the observed concentration of $CO_2$ from the K30 sensor to a value approximately that of the concentration determined from the calibration-corrected FGGA measurements. Generally, a multivariate linear regression is of the form shown in Eq. (1):

$$y = a_1 x_1 + a_2 x_2 + \cdots a_n x_n + \varepsilon_n \qquad (1)$$

In this case, the measured value $y$ is influenced by: the 'true' $CO_2$ value (taken as the value from the LGR FGGA instrument), pressure, and other environmental variables as the dependent variables $x_1$, $x_2$, $x_n$, respectively. A multivariate regression analysis can then be used to find the corresponding coefficients. In addition, in order to better identify the contribution from each individual factor, the data was also analyzed in a successive regression analysis, as described below.

### 5.1 Successive regression method

Each individual K30 sensor's original observed $CO_2$ dataset is first regressed to the LGR FGGA dry $CO_2$ dataset. This regression accounts for the traditional zero and span corrections made during an instrument calibration. The calibration curve of one K30 for just zero and span is shown in Fig. 5. But to include biases due to environmental factors, then the residual, epsilon ($\varepsilon$), is calculated in Eq. (2) as:

$$\varepsilon = y - ax - b \qquad (2)$$

where in this instance $x$, the independent variable, is the FGGA dataset and $y$, the dependent variable, is the K30 dataset.




This process is repeated for each environmental variable pressure (P), temperature (T), and water vapor (q), where $(P,T,q)$ is the independent variable, $x$, and the $\varepsilon$ from the previous step is the dependent variable, $y$. This linear regression method leads to eight correction coefficients, of the form $a_n$ and $b_n$, where $n$ is from 0 to 3 representing each of the independent variables included in the regression. These coefficients can then

be used in Eq. (3) along with the environmental variables to correct K30 $CO_2$ observations for environmental influences.

$$y_{corrected} = \frac{y - b_0 - (a_1 x_1 + b_1) - (a_2 x_2 + b_2) - \cdots - (a_n x_n + b_n)}{a_0} \tag{3}$$

For one typical K30, the initial standard deviation of the difference between the K30 and FGGA, the RMSE

of the data was 6.68 ppm. Using the cumulative univariate regression method described above for the entire evaluation period, the RMSE decreased after each step. After the span and offset regression, it dropped significantly to 3.49 ppm. Then after correcting for atmospheric pressure, the RMSE dropped even lower to 3.01 ppm. Furthermore, including air temperature and water vapor mixing ratio dropped the RMSE to 2.97 ppm and 2.25 ppm respectively. Therefore, using the successive regression method, the RMSE of the

observed difference dropped from 6.68 ppm to 2.25 ppm, a reduction of the error by nearly a factor of three. Fig. 6 shows the results and scatter plots for each step of the correction for this K30; Fig. 7 shows a difference plot at each step for this same K30 unit. Similar results were observed for each K30 sensor evaluated and a summary can be found in Table 1.

**5.2 Multivariate linear regression method**

Alternatively, a multivariate linear regression statistical method can be used to calculate the regression coefficients for each K30 sensor. This results in five correction coefficients $a_n$ and $b$ where $n$ represents each independent variable, the dry $CO_2$ from the LGR FGGA, pressure $P$, temperature $T$, and water vapor mixing ratio $q$. Like the successive method above, these coefficients can be used in Eq. (4) along with the original K30 data, $y$, and the environmental variables to predict the true $CO_2$ concentration observed.

$$y_{corrected} = \frac{y - b - (a_1 x_1) - (a_2 x_2) - \cdots - (a_n x_n)}{a_0} \tag{4}$$

Using the multivariate regression function provided by Python-SciPy-Stats (Jones et al., 2001), differences from the FGGA of the same K30 described in Sect. 5.1 were reduced to an RMSE of 1.89 ppm, slightly

better than the iterative method. This consistently better performance from the multivariate method is shown in the other K30 sensors evaluated. Figure 8 shows the final results of the multivariate regression for the same K30 as in Fig. 6 and Fig. 7, as well as the difference between the corrected K30 dataset and the



FGGA. As with the univariate method, similar results were observed from each K30 sensor evaluated and a summary can also be found in Table 1.

## 6 Discussion

### 6.1 Time averaging

There are two observations to note based on the evaluation and analysis. First, both before and after the multivariate regressions, there are frequent shifts in the sign of the difference between each K30 and the FGGA; these sudden changes occur at or around sunrise most days. Because of the rapid change in atmospheric $CO_2$ concentration at this time, the ambient calibration chamber may not be well mixed during this time period. Each K30 is located in a slightly different location in the ambient calibration chamber, and are all approximately 1 to 2 meters away from the FGGA inlet. This effect, combined with the different response time of the K30s compared to the FGGA, can lead to dramatic differences between what each K30 observes and what the FGGA observes at the same timestamp for a short period of time each day.

Atmospheric inversion methods often use hourly averaged data from tower observations (McKain et al., 2012; Bréon et al., 2015; Lauvaux et al., 2016), so after the multivariate regression was applied, the K30 and FGGA datasets were further averaged to 10 minute and hourly datasets. The average RMSE for the six K30s with the one-minute data is 2.32 ppm, 2.04 ppm for 10-minute averages, and 1.87 ppm for hourly-averaged data. Throughout this analysis period, one of the six K30s evaluated performed consistently worse than the others, and after removing it from the averages, the RMSE values dropped to 1.90 ppm, 1.66 ppm, and 1.48 ppm, respectively. Thus, by using hourly averages and discarding underperforming sensors, the average RMSE of the difference between the LGR FGGA and a K30 NDIR sensor can be reduced to approximately 1.5 ppm.

### 6.2 Regression period

The RMSE described above and in Table 1 are for regressions calculated over the entire experiment period of approximately four weeks. One goal of this work is to evaluate these sensors individually quickly so that they can be used in scientific applications. In Fig. 9 the average RMSE calculated over the entire month of all six K30s is plotted with respect to the number of days used in the multivariate regression from Sect. 5.2. While the RMSE is generally minimized with increasing regression length, after a regression period of just a few days, the RMSE drops significantly from its initial values. Once a few diurnal cycles of varying amplitude have been incorporated, as well as the synoptic scale variations in the atmosphere (with a time scale of around one week), the regression stabilizes. Thus, a regression length of around two weeks is





recommended to maximize correction while minimizing the required amount of time the sensor needs to run concurrently with the FGGA.

In Fig. 10, a multivariate regression is applied to the same K30 as described in the aforementioned sections and shown in Figs. 6, 7 and 8, but the coefficients are only calculated for the first 15 days. The change in the RMSE between the two regressions is 0.25 ppm, going from 1.89 ppm when using all data points to 2.14 ppm when using only approximately the first half. This small, but not insignificant change is most likely attributed to the fact that during the first half of the evaluation period, the ambient $CO_2$ concentrations do not vary significantly, especially relative to the second half, where both the minimum and maximum values occur. In fact, when instead regressing for the last 15 days of the period, the RMSE is 1.94 ppm, a difference of only 0.05 ppm. So as stated above, the diurnal cycles act as a range of calibration points, but values above and below what is included in the regression period may cause the corrected data to still have large errors during these periods, increasing the RMSE for the entire evaluation cycle. Based on these results, it is reasonable to assume that there is either no noticeable baseline drift or that it is assumed to be linear and removed by the multivariate regression in the sensors observed on the weekly to monthly timescales. The longer-term drift of the sensors for periods greater than one month is not known at this time, however, and would require a longer evaluation period of at least six months.

## 7. Conclusions and future work

In the future, further analysis will be performed evaluating the K30 as well as other low-cost $CO_2$ sensors in a laboratory setting with controlled temperature, pressure and relative humidity. A Picarro cavity ring-down spectroscopy greenhouse gas analyzer will be used as a high-precision control and the various instruments will be subjected to ambient air as well as periodic reference gases. From this lab analysis, we hope to determine the theoretical maximum performance of these sensors in a controlled environment. This subsequent study will additionally attempt to quantify any long-term drift over the course of multiple months.

The K30 is a small, low-cost NDIR $CO_2$ sensor designed for industrial OEM applications. Each of the sensors tested falls within the manufacturer's stated accuracy range of ±30 ppm ±3 % of the reading when compared to a high-precision CEAS analyzer, but these ranges are not particularly useful for scientific applications aimed at measuring ambient atmospheric $CO_2$. If these sensors are individually calibrated, selected for stability, and corrected for sensitivity to temperature, pressure, and RH, the practical error of these sensors is less than five parts per million, or approximately 1 % of the observed value. The final



RMSE of the six K30 ranged between 1.6 ppm and 4.4 ppm for 60 s averaging times. Averaging for 200 s further reduces the noise by about 30 %, but longer times produce no lower noise. With errors in this range, these instruments could be used in a variety of scientific applications, including observations at high spatial density to better represent the range and distribution of an urban or natural region's $CO_2$ concentration.

*Acknowledgements*. We acknowledge support for this project from the FLAGG-MD grant from NIST's Greenhouse Gas and Climate Science Measurements program (Cooperative Agreement #70NANB14H333). The authors wish to thank the undergraduate and graduate students at the University of Maryland who helped with this analysis. Additionally, we would like to thank all of the members of the NIST Greenhouse Gas and Climate Science Measurements program including Subhomoy Ghosh, Israel
Lopez-Coto, Kimberly Mueller, Kuldeep Prasad, James Whetstone, and Tamae Wong for their help.





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





**Table 1.** Root mean square error in ppm between the CEAS LGR FGGA and each K30 NDIR sensor's one-minute averaged data for: the original dataset before correction, at each step of the successive regression correction (correcting for 1. zero/span, 2. atmospheric pressure, 3. temperature, and 4. water vapor mixing ratio), and after the multivariate regression correction. Each value shown is for a regression calculated using data from the entire evaluation period.

|          | Original | Zero/Span | Pressure | Temp | q (final) | Multivariate |
|----------|----------|-----------|----------|------|-----------|--------------|
| K30 # 1  | 6.68     | 3.49      | 3.01     | 2.97 | 2.25      | 1.89         |
| K30 # 2  | 5.13     | 3.54      | 2.48     | 2.47 | 2.04      | 1.80         |
| K30 # 3  | 11.24    | 6.13      | 5.33     | 5.13 | 4.60      | 4.42         |
| K30 # 4  | 21.01    | 3.67      | 2.61     | 2.49 | 1.88      | 1.58         |
| K30 # 5  | 8.08     | 3.74      | 2.91     | 2.87 | 2.31      | 2.04         |
| K30 # 6  | 15.04    | 4.85      | 3.60     | 3.48 | 2.69      | 2.19         |



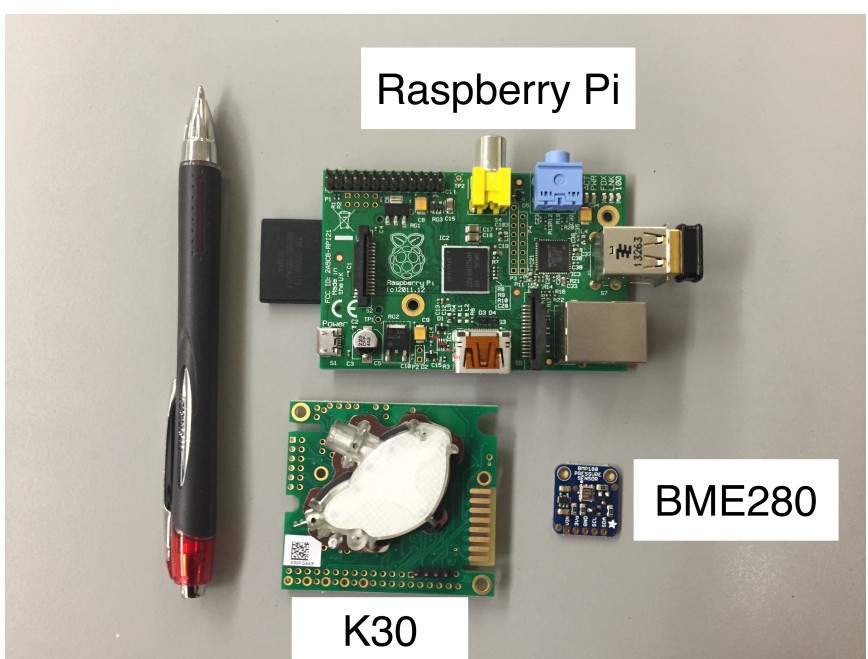

**Figure 1.** Photograph of a Raspberry Pi computer (top), a SenseAir K30 (NDIR) $CO_2$ sensor (bottom center), a Bosch BME280 temperature and pressure sensor (bottom right), and a ballpoint pen for size reference.



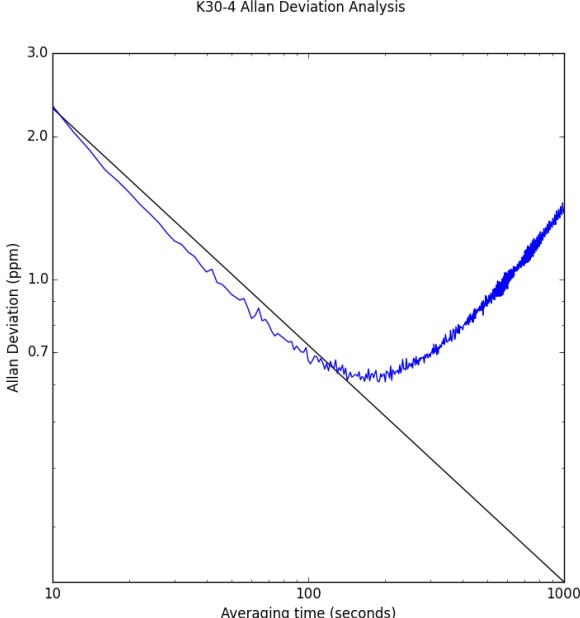

**Figure 2.** Allan variance analysis for an NDIR (K30) $CO_2$ sensor when introduced to breathing air from a high-pressure cylinder of a constant and known $CO_2$ concentration. Averaging times between 10 and 1,000
5    seconds are shown. The black line (slope -0.5) shows where the noise is white or Gaussian. Averaging times greater than about 200 s produce no improvement.

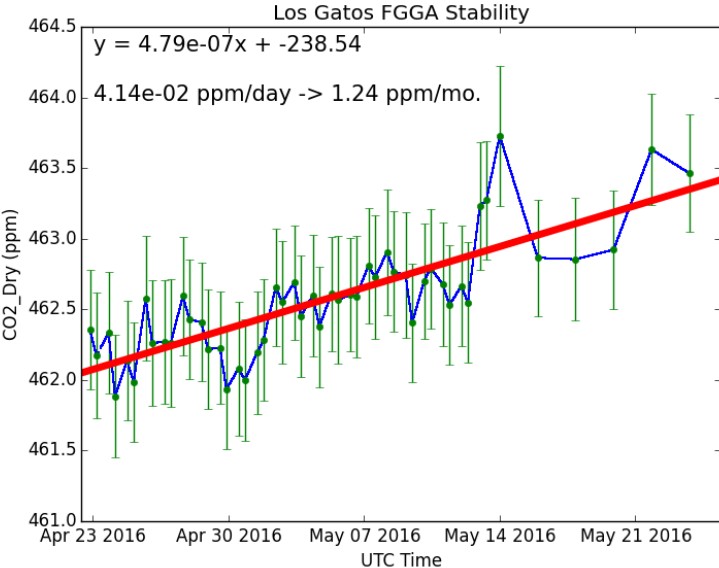

**Figure 3.** Stability of the Los Gatos Fast Greenhouse Gas Analyzer shown over a 30-day period. Excess
10    breathing air with a fixed $CO_2$ concentration was introduced periodically using a mass flow controller.





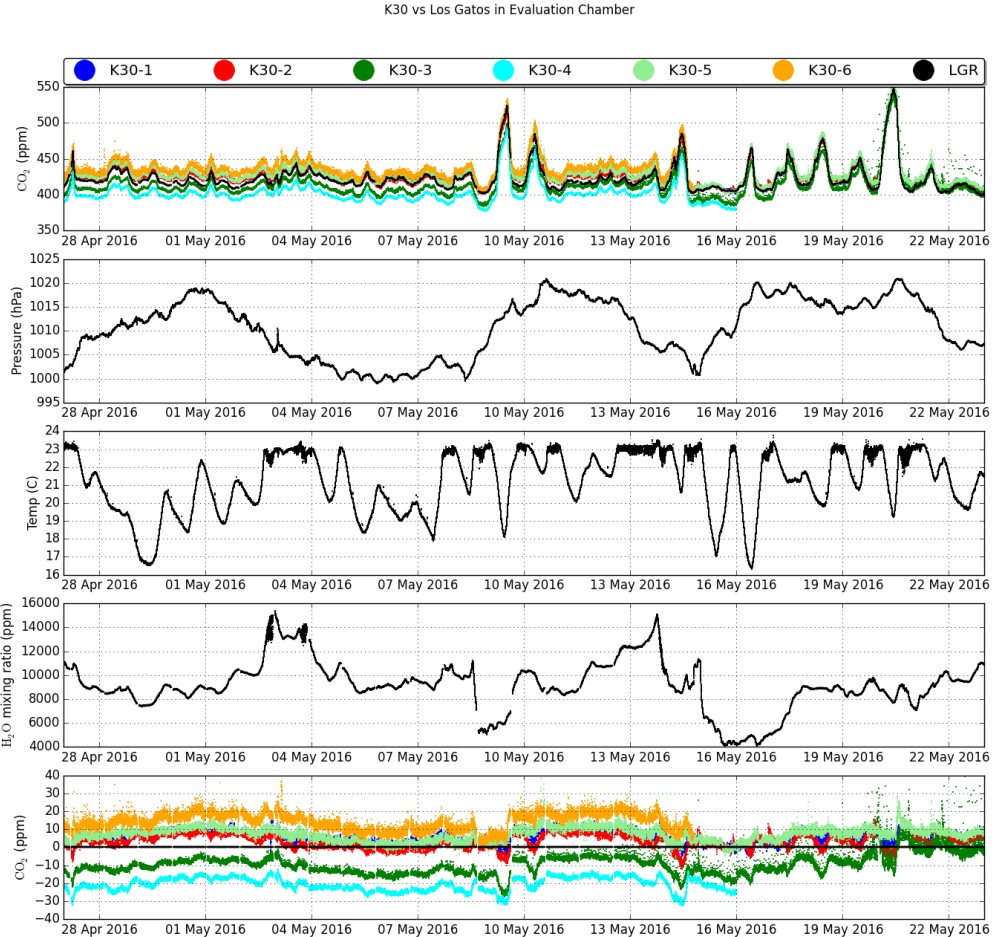

**Figure 4.** Continuous time series data during the evaluation experiment. Top panel: $CO_2$ observed by six K30 sensors as well as the Los Gatos Research Fast Greenhouse Gas Analyzer. Middle panels: observed atmospheric pressure, temperature, and water vapor mixing ratio, respectively. Bottom panel: difference of each K30 from the Los Gatos instrument.





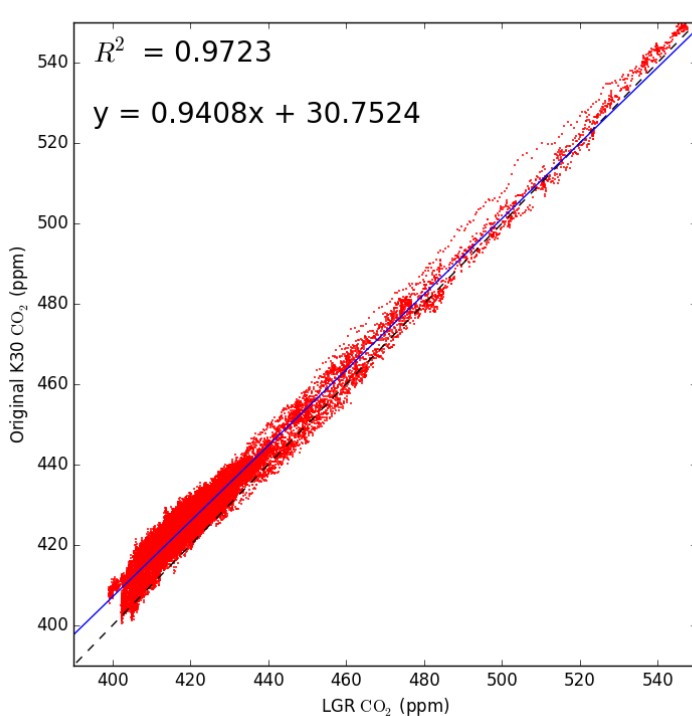

**Figure 5.** Calibration curve of K30-1 vs LGR FGGA without any environmental correction, only span and zero offset are corrected. Solid line is the best fit; dashes represent the 1:1 line




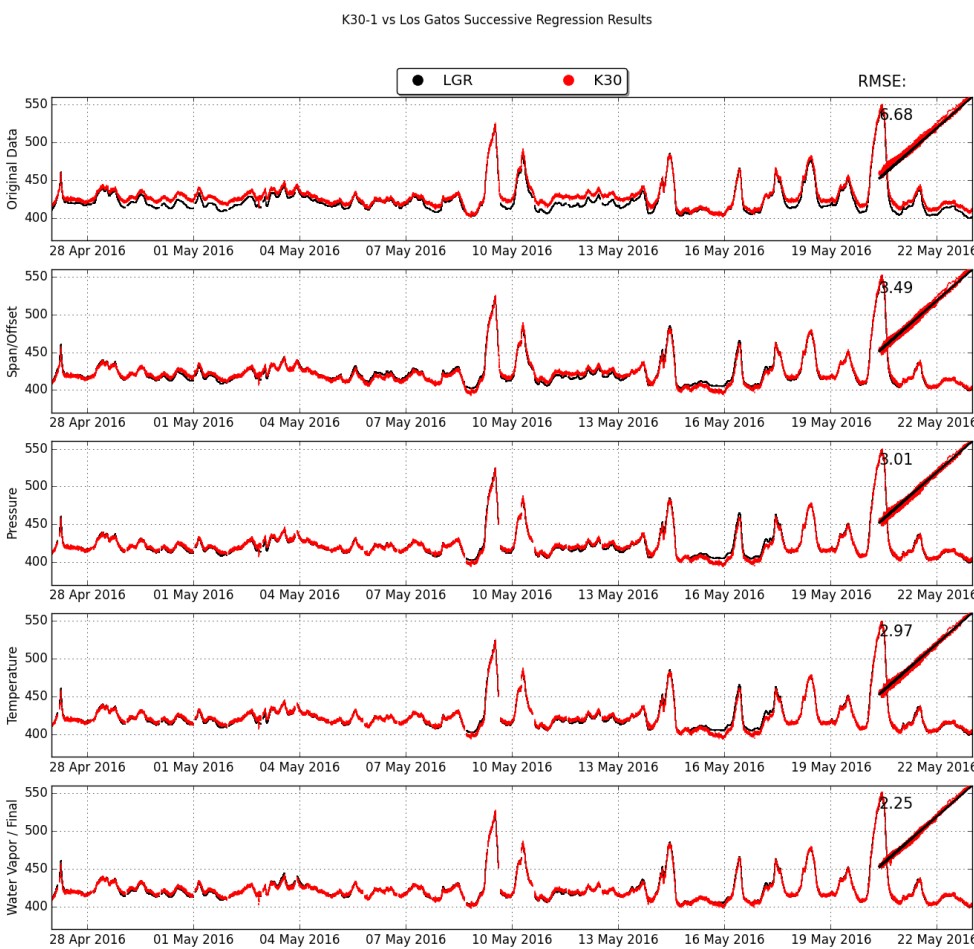

**Figure 6.** A continuous time series as well as scatter plots for K30 #1 compared to the LGR instrument during each step of the successive regression described in Sect. 5.1. Cumulative, in order from top to bottom: the original dataset, after correcting for span and offset, after correcting for pressure, after correcting for temperature, and finally, after correcting for water vapor. The root mean square error (RMSE) of the K30 data compared to the LGR FGGA at each step is annotated to the upper left of the scatter plot. This regression contains all data points observed in the evaluation period.


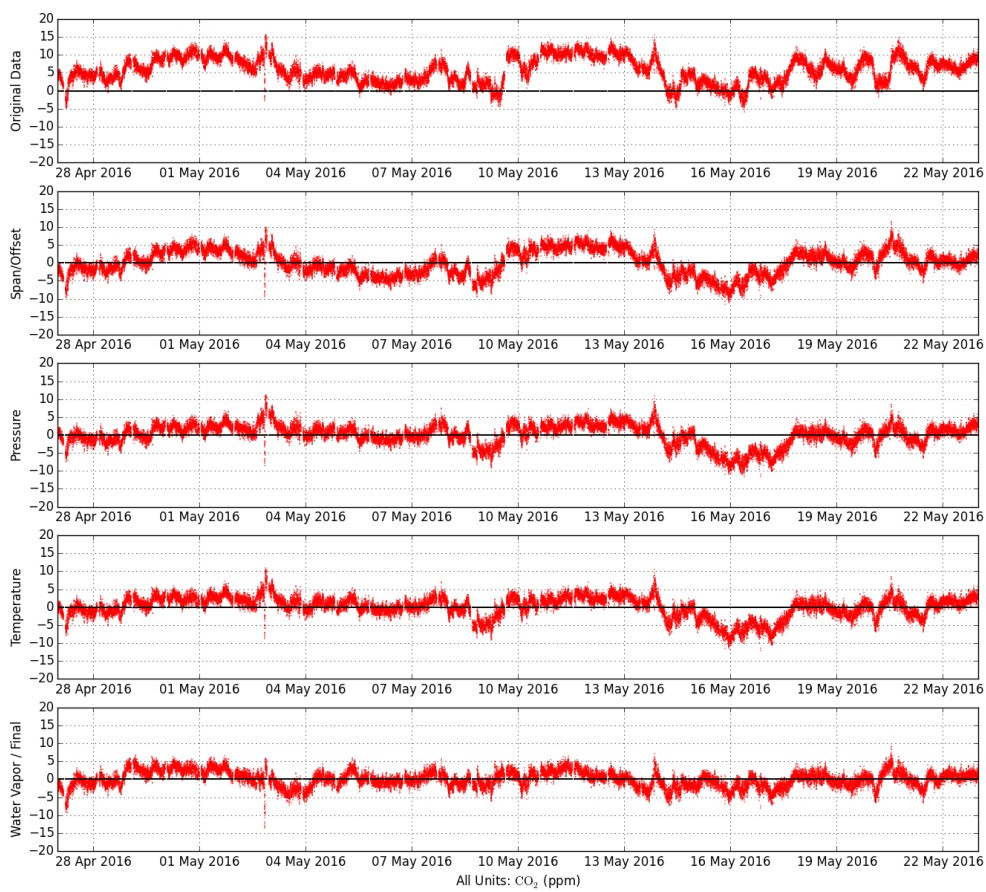

**Figure 7.** Difference plots for K30 #1 compared to the LGR FGGA during each step of the successive regression described in Sect. 5.1 and shown in Fig. 6. Cumulative, in order from top to bottom: the original dataset, after correcting for span and offset, after correcting for pressure, after correcting for temperature, and finally, after correcting for water vapor.





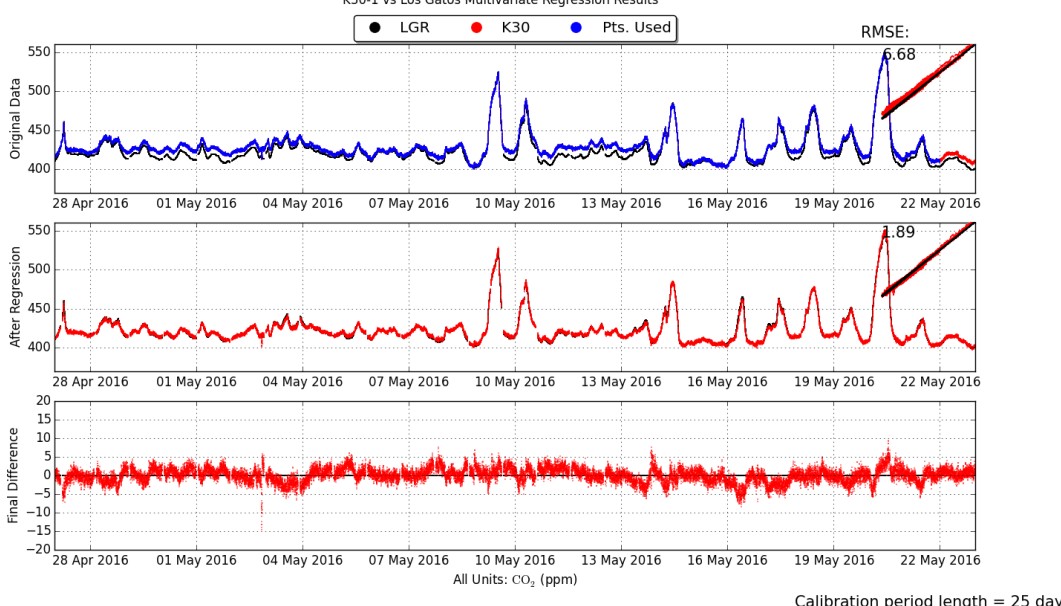

**Figure 8.** A continuous time series as well as scatter plots for K30 #1 compared to the LGR FGGA for the multivariate regression described in Sect. 5.2. Top panel: the original data, middle panel: final time series after correction, and the bottom panel: difference plot between the corrected K30 dataset and the original FGGA dataset. The root mean square error (RMSE) of the K30 data compared to the FGGA before and after the regression is annotated to the upper left of the scatter plot. The blue data points are used in the regression, which cover the first 25 days of the evaluation period.





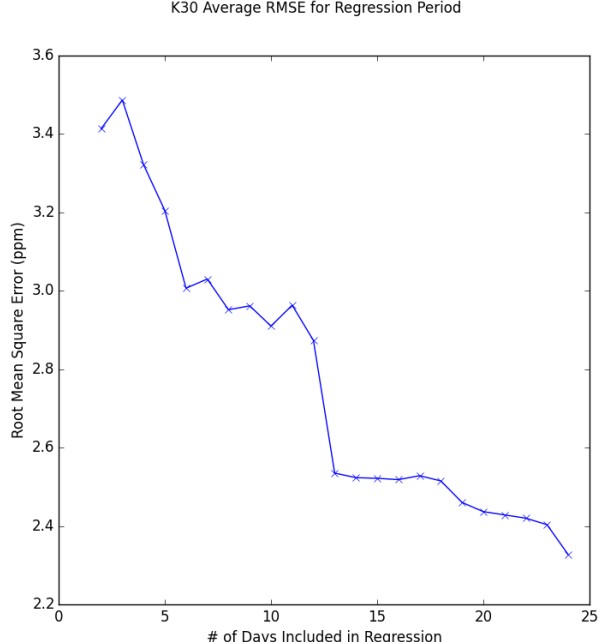

**Figure 9.** The average RMSE of all six K30 NDIR sensors when compared to the LGR FGGA over the entire experiment as a function of how many days the regression analysis was performed.





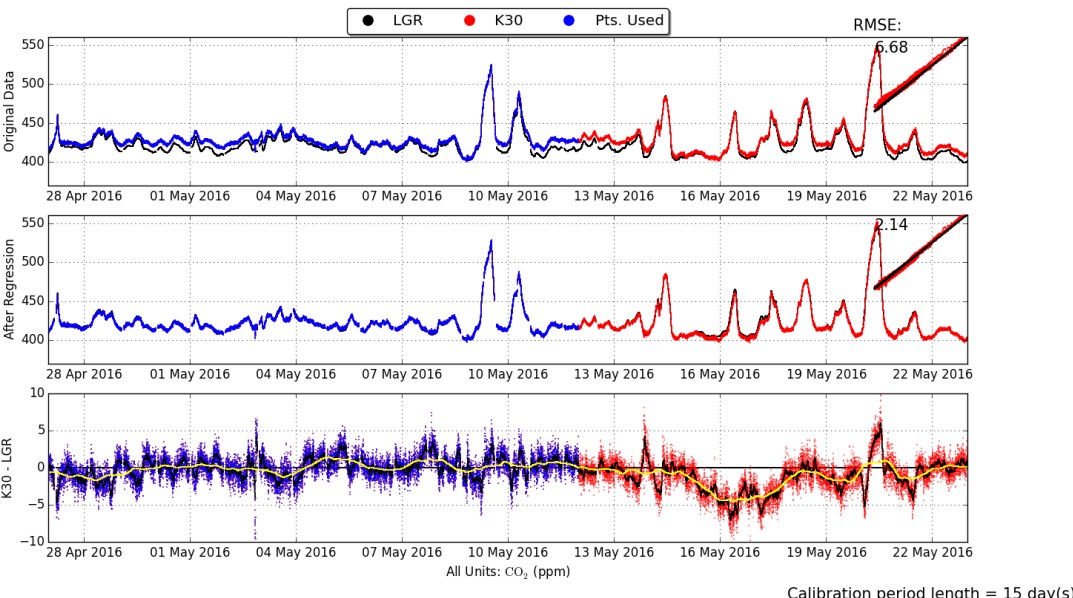

**Figure 10**. As depicted in Fig. 8, a continuous time series as well as scatter plots for K30 #1 compared to
the LGR FGGA for the multivariate regression described in Sect. 5.2. Top panel: the original data, middle
panel: final time series after correction, and the bottom panel: difference plot between the corrected K30
dataset and the original FGGA dataset. However, this regression only includes the first 15 days of data (in
blue) to compute the correction coefficients. The difference plot (bottom) also shows running means for 10
minute (black) and hourly (yellow) averages.