# Peer review of "Evaluation and environmental correction of ambient CO2 measurements from a low-cost NDIR sensor"

_Atmospheric Measurement Techniques, 2016_

## Referee Comment (RC1) · Anonymous Referee #2 · 6 Feb 2017

General comments

This study characterizes the performance (accuracy, precision, drift) of one type of low-cost CO2 sensor for ambient air measurements. The paper describes an experiment where duplicate sensors are used to make continuous ambient measurements in an environment with conditions that change slowly over time. Ambient pressure, temperature, and humidity are monitored simultaneously and used to derive empirical corrections for individual sensors, which significantly improve the accuracy of the final datasets. The overall experiment is well-designed and the analysis of the resulting data is sound.

The paper topic is highly relevant and potentially useful to the broader atmospheric measurement community, but currently falls short of that potential. There are several

additional experiments and analyses one could imagine that would fit in the same paper and would improve the scope and significance, such as: (i) test whether a unique correction is needed for each unit and what the uncertainty would be if a generalized correction were applied, (ii) demonstrate an experiment which would allow correction factors to be rapidly derived in the lab, (iii) test the K30 in a real-world environment, (iv) test the K30 for long-term drift. At a minimum, it appears that the authors can use the existing dataset to address point (i).

Specific comments

Los Gatos Instrument

You use the LGR dataset as a control, but I am concerned that there could be large uncertainty associated with the LGR water correction. Do you know the accuracy of the LGR water correction? I have never seen an assessment of it.

You calibrate the instrument with two points by assuming a linear fit. How do you know the true instrument response is linear?

How did you decide to measure the tank at 23 and 47 hour intervals? How do you know that significant drift did not occur over shorter intervals?

Why do you need to measure the tank for such long time periods (10-60 minutes)? Does it take that long for the measurement to equilibrate? If you are using the proper materials in your plumbing, the measurement should equilibrate in a matter of seconds to minutes. If it is taking a long time for the $CO_2$ signal to equilibrate, that suggests that $CO_2$ may be absorbing/desorbing onto the walls in your plumbing.

What is the purpose of the Dasibi calibrator? Did you have to dilute the tank air to get ambient values?

Figure 3: If you take the linear trend out, are the remaining variations related to a physical parameter such as temperature (ambient or cell)?

Your drift correction technique of fitting a line to each subsequent pair of calibration points will introduce discontinuities into the corrected dataset that do not represent the real-world. It would be better to fit a smoothed curve (captures short-term drift) or a single linear fit (captures the long term drift).

When describing the differences between the K30 and LGR, you say that the LGR cavity temperature and pressure are relatively well controlled. Please give some numbers to give us a sense of how well controlled they are.

You average the datasets into 1-minute bins based partially on the Allan variance results for the K30. Did you also do an Allan variance for the LGR?

K30 Sensor Performance and Evaluation

Sect 2.1 – Did you compute an Allan variance for more than one sensor? Do they all perform similarly?

Figure 4 – $CO_2$ traces show periods of higher noise on some sensors (e.g. K30-3 during the second half of the time period shown). In particular, I am wondering about the smattering of points that appear as outliers. In these cases, are the sensors still meeting the manufacturer's specification of +/- 30 ppm? Is there evidence that a sensor's precision can diminish over time?

At the beginning of section 5, you state that $CO_2$ measurement differences are correlated with environmental variables, but you have not demonstrated the correlation. Can you show some scatter plots?

Before doing an empirical fit to the environmental parameters, it would seem sensible to account for the dilution of the $CO_2$ mixing ratio in humid air. See section 2 of Shusterman et al. 2016 for an example.

Table 1 – Are all of the regression coefficients significant? Which parameter leads to the biggest improvement and which leads to the smallest improvement?

Section 6.1 – Do you find that the K30 sensors that were closer to the LGR inlet have shorter lag times relative to the LGR response? You should try computing cross-correlation functions for each K30 against the LGR to improve the time-matching of the different time series.

Significant figures - Most of the performance metrics stated for the K30 sensors in units of ppm $CO_2$ are given with two decimal places, yet you state the K30 measurement resolution is only 1 ppm.

Figures 5,6,7,8,10 are all shown for K30 #1, which, from Figure 4, appears to be one of the best performing sensors. I would be curious to see a residual plot for sensor #3, 5 or 6.

You state that one goal of this work is to understand how correction coefficients can be derived quickly. Wouldn't it be more efficient to design a controlled experiment where controlling variables are deliberately varied across the full range of operating conditions?

Section 7 – In future work, you aspire to characterize the sensors' maximum performance in a controlled environment. Yet, if the big-picture goal is to use these sensors is to generate science quality ambient air measurements, I believe a more worthy goal would be to characterize their minimum performance in an uncontrolled environment.

Technical comments

Title – I don't think "enhancement" is the right word. How about something like "Evaluation and correction of $CO_2$ measurement in ambient air from a low-cost sensor"

Abstract – The quantities reported have different numbers of significant figures. These should be uniform and reflect the precision of the measurement.

Pg 1, Ln 25 – "dry air" is used twice in this sentence.

Pg 1, Ln 28 – The WMO compatibility goal is a goal, but is not always achieved, and

certainly not for historical measurements.

Pg 1, Ln 29 – Suggest: '...to collect samples, which are subsequently transported'

Pg 1, Ln 30 – You mention two expensive types of measurements – flasks and Picarros, but you do not mention moderately-priced analyzers from LiCor and Los Gatos, which are used at many research-grade monitoring sites.

Pg 2, Ln 1 – Be consistent about whether you spell out carbon dioxide or use the abbreviation.

Pg 2, Ln 2 – Suggest paragraph break at "Recent research"

Pg 2, Ln 8 – Is 8-12 sites typical? There are ∼5 sites in Boston, SLC, and Paris.

Pg 2, Ln 9 – You say that more dense observations, even with larger uncertainties, yield better inversion constraints, but this is all relative and depends on the inversion setup/goals. See Turner et al., 2016, ACP for an exploration of the tradeoffs.

Pg 2, Ln 14 – Suggest deleting "however"

Pg 2, Ln 16 – Suggest changing the phrasing to: "Recent evaluations and implementations of new low-cost sensors demonstrate their promise for ambient air monitoring."

Pg 2, Ln 26 – Can you give some numbers to scope what you mean by "reasonably accurate"?

Pg 3, Ln 7 – Suggest: "The K30 sensor moduel from SenseAir (Sweden) is the loc-cost NDIR CO2 sensor that was tested for this study".

Pg 3, Ln 10 – Suggest deleting "given as"

Pg 3, Ln 13 – Suggest: "The K30 was chosen not only because it has the highest manufacturer-specified accuracy, but also because initial testing showed reliability and consistency with higher-quality observations."

Pg 3, Ln 17 – You should give the units (relative humidity) for the 3% and 0.008%

Interactive
comment
quantities.

Pg 3, Ln 24 – "less than one percent" "< 1%"

Pg 4, Ln 5 – Another difference between the two analyzers could be their sensitivity to the isotopes of $CO_2$.

Pg 4, Ln 29 – Can you briefly describe what you mean by "various complications"?

Pg 5, Ln 1 – My understanding is that you merged datasets by their timestamps. Did you have to do something to keep the clocks synchronized?

Pg 5, Ln 13 – "longer averaging times do not reduce the noise"

Pg 7, Ln 21 – Suggest deleting "However".

Pg 7, Ln 22 – Is the statement about each K30 meeting the manufacturer's uncertainty specification in regards to the raw (2-second) data or 1-minute averages? Please clarify in the text.

Pg 10, Ln 21 – suggest: ". . . and 1.48 ppm, for 1-minute, 10-minute, and hourly averages, respectively.

Pg 10, Ln 26 – suggest: "One goal of this work is to develop a methodology to evaluate individual sensors quickly. . ."

Pg 11, Ln 32 – "less than five parts per million" "< 5 ppm"

Figure 1 – A ballpoint pen is included in the picture for size reference. A ruler instead of a pen would be more useful.

Figure 2 – What was the $CO_2$ concentration of the tank used?

Figure 4 – State the time interval of the data shown. I can't tell if this is raw 2-second data or 1-minute averages.

Figure 8 – I don't understand the difference between the red and blue points.

[Figure]

Figure 9 – Can you put error bars on each point for the y-variable?

[Figure]

---

## Referee Comment (RC2) · Anonymous Referee #1 · 7 Feb 2017

The presented article address the application of low-cost NDIR CO2 sensors for urban measurement networks aimed at assessment of CO2 fluxes over an urban areas using inverse modelling technique. While such sensors has not enough precision in case of background studies, their application in urban areas where amplitude of atmospheric CO2 mixing ratios is order of magnitude higher is possible. These circumstances make the presented study an important contribution to the construction of such measurement networks. Authors focused on evaluation of several copies of SenseAir K30 NDIR CO2 sensor. Authors demonstrated that all the sensors fulfill the technical specification of manufacturer however this specification is not enough in above application. A series of long term measurements showed that application of correction factors determined by two statistical approaches, subsequent univariate and multivariate linear regression analysis significantly improve the sensor performance.

Detailed comments:

p.1 l.15: there is no info on RMS of research-grade analyzer used by authors which is compared to low-cost

p.4 l.3 Why authors decided to use such narrow range of CO2 mixing ratios. In real urban environment values close to 500 ppm or more are frequently observed.

p.6. L23 it is not clear why the calibration strategy has been changed during the experiment. Why some standards were flushed for 10 min and other for one hour?

p.11 l.19 Authors decided to use natural synoptic variability to perform a regression analysis aimed at determination of correction factors taking into account the influence of temperature, humidity and pressure variability of NDIR sensors on CO2 measurements. Such procedure requires long time and is depended on existing natural variability. To standardize and shorten the procedure maybe a construction of special environmentally controlled chamber should be taken into account?
* * *

---

## Author Comment (AC2) · 5 Apr 2017

**Response to Anonymous Referee #1**

Responses in blue

The presented article address the application of low-cost NDIR CO2 sensors for urban measurement networks aimed at assessment of CO2 fluxes over an urban areas using inverse modelling technique. While such sensors has not enough precision in case of background studies, their application in urban areas where amplitude of atmospheric CO2 mixing ratios is order of magnitude higher is possible. These circumstances make the presented study an important contribution to the construction of such measurement networks. Authors focused on evaluation of several copies of SenseAir K30 NDIR CO2 sensor. Authors demonstrated that all the sensors fulfill the technical specification of manufacturer however this specification is not enough in above application. A series of long term measurements showed that application of correction factors determined by two statistical approaches, subsequent univariate and multivariate linear regression analysis significantly improve the sensor performance.

Thank you for your time in reviewing our manuscript, we appreciate your acknowledgement that this work has the potential to help improve the constraint of $CO_2$ emissions from urban areas.

Detailed comments:

p.1 l.15: there is no info on RMS of research-grade analyzer used by authors which is compared to low-cost

The LGR analyzer was calibrated as described in section 3 with two NIST-traceable gas standards as well as evaluated for drift over the experiment using a tank of breathing air. The Allan variance was computed for the LGR and determined that ~100 seconds was the optimum averaging time for noise. Additionally, after correcting for the drift, the noise in the LGR for the breathing air tank was ±0.3 ppm for 2-second data, which is within the manufacturer's specifications. This 1-sigma standard deviation has been added to the manuscript.

p.4 l.3 Why authors decided to use such narrow range of CO2 mixing ratios. In real urban environment values close to 500 ppm or more are frequently observed.

The ambient experiment was conducted in College Park, Maryland, approximately 12 km from the central business district of Washington, DC, and a couple of the nights the concentration did exceed 50 0ppm, but was usually below 450 ppm. These two tanks were used for calibration because they were readily available from other experiments at the University of Maryland, where the application is to calibrate boundary layer observations on aircraft, where the concentrations are generally lower. In addition, three breath air cylinders with higher $CO_2$ mole fractions of

449.73, 486.53, and 516.41ppm (all NIST-traceable) were also used to calibrate the LGR.

p.6. L23 it is not clear why the calibration strategy has been changed during the experiment. Why some standards were flushed for 10 min and other for one hour?

We were initially unsure how long the LGR needed to equilibrate, and wanted an idea of the variability/stability at the fixed concentration, which is why we initially ran the calibration gas for 60 minutes. Looking at the raw data, it takes somewhere on the order of 90-120 seconds to fully equilibrate. It was switched later to conserve the breathing air tank.

p.11 l.19 Authors decided to use natural synoptic variability to perform a regression analysis aimed at determination of correction factors taking into account the influence of temperature, humidity and pressure variability of NDIR sensors on CO2 measurements. Such procedure requires long time and is depended on existing natural variability. To standardize and shorten the procedure maybe a construction of special environmentally controlled chamber should be taken into account?

In an ideal situation, this is what should be done, but is impractical for a large number of sensors. To cycle temperature, pressure, and relative humidity throughout typical ambient ranges is not difficult for one instrument, but for several requires a large enough chamber, and needs to be set up to be autonomous, otherwise if someone is manually controlling these parameters for days/weeks, the low-cost aspect of these instruments becomes much more labor intensive. An additional point that was added to the revised manuscript as suggested by another reviewer, is to see if each sensor has to be individually evaluated. Since it appears that a uniform set of regression coefficients does not work, each sensor requires a 15-30 day evaluation period and a chamber would need to be large enough to contain at least 6, but preferably 10 or more to be viable. Using the natural variability to calibrate the sensors allowed the experiment to be conducted without supervision once it was set up.

---

## Author Comment (AC1)

**Response to Anonymous Referee #2**

Responses in blue

General comments

This study characterizes the performance (accuracy, precision, drift) of one type of low-cost $CO_2$ sensor for ambient air measurements. The paper describes an experiment where duplicate sensors are used to make continuous ambient measurements in an environment with conditions that change slowly over time. Ambient pressure, temperature, and humidity are monitored simultaneously and used to derive empirical corrections for individual sensors, which significantly improve the accuracy of the final datasets. The overall experiment is well-designed and the analysis of the resulting data is sound.

The paper topic is highly relevant and potentially useful to the broader atmospheric measurement community, but currently falls short of that potential. There are several additional experiments and analyses one could imagine that would fit in the same paper and would improve the scope and significance, such as: (i) test whether a unique correction is needed for each unit and what the uncertainty would be if a generalized correction were applied, (ii) demonstrate an experiment which would allow correction factors to be rapidly derived in the lab, (iii) test the K30 in a real-world environment, (iv) test the K30 for long-term drift. At a minimum, it appears that the authors can use the existing dataset to address point (i).

First off, thank you very much for your thorough review of our paper and for the helpful comments and suggestions. For your general comments:

(i) This additional analysis has been added to Section 6 as was suggested, but to summarize, yes a unique correction is required. By taking the average coefficients and intercepts computed by the multivariate regression of the five best performing K30s, all six sensors had higher RMSEs than with independent coefficients, which is expected. But at a minimum, the RMSE doubled with in some cases actually became worse than with no correction at all. Thus, instead of 1.5-3 ppm RMSE after correction, it can range from 3 to 24 ppm for 1 minute data.

(ii) The main reason why we did an ambient calibration, and the main concern with doing a laboratory correction is the labor and equipment required to do this. To cycle temperature, pressure, and relative humidity throughout typical ambient ranges is not difficult for one instrument, but for several requires a large enough chamber, and needs to be set up to be autonomous, otherwise if someone is manually controlling these parameters for days/weeks, the low-cost aspect of these instruments becomes

much more labor intensive. At this time, we did not have the resources or an environmental chamber available to conduct this experiment. This is something we hope to do with future work, as mentioned in the conclusions.

(iii)    The experiment described in the manuscript is a quasi-real-world environment, as we used ambient air and ambient variations in the environmental variables, but the main difference was the sensors were not outside in direct sunlight or exposed to weather. This is something we hope to do in the future, but we wanted to evaluate them without these engineering concerns initially.

(iv)    This is also something we hope to achieve with future work (as mentioned in the conclusion portion of the paper). While it would have been nice to include long-term drift in this manuscript, that would require 6-12 months of data with either a continuous gas analyzer as reference or calibration gas introduced periodically. We believe it is sufficient to publish these initial results characterizing month-long drift, as the results will be useful to others working with similar sensors. We will include an evaluation of long-term drift of this sensor and others in a future manuscript.

Specific comments

Los Gatos Instrument

You use the LGR dataset as a control, but I am concerned that there could be large uncertainty associated with the LGR water correction. Do you know the accuracy of the LGR water correction? I have never seen an assessment of it.

Yang et al. 2016 describes a comparison between a Picarro dried with Nafion and a LGR for flux measurements and found an $R^2$ of 0.99 for $CO_2$. While the correction is not perfect, it seems sufficient for our purposes, particularly that we are attempting to determine the accuracy and precision of the NDIR sensors relative to the LGR, not to the absolute concentration.

Additionally, previously this instrument's water vapor correction was evaluated, but was focused on $CH_4$ by adding different moisture into the sample line with a calibration gas flowing. The measured [$CH_4\_dry$] is constant at different RH values. This experiment was not designed for $CO_2$ assessment because the calibration gas went through a bubbler at different flow rates and some $CO_2$ dissolved into the water.

You calibrate the instrument with two points by assuming a linear fit. How do you know the true instrument response is linear?

A previous calibration for this instrument was done with five NIST-traceable standards ranging from 369.19 to 516.41 ppm which showed its linearity. This has been clarified in the text.

How did you decide to measure the tank at 23 and 47 hour intervals? How do you know that significant drift did not occur over shorter intervals?

We wanted calibrations to occur at different times throughout the day, thus the 23 and 47 hour intervals, and was not performed more often in an attempt to have enough calibration gas for the entire period. As described below, we were unsure how long we would need to equilibrate/get an idea of the repeatability of the LGR instrument.

Why do you need to measure the tank for such long time periods (10-60 minutes)? Does it take that long for the measurement to equilibrate? If you are using the proper materials in your plumbing, the measurement should equilibrate in a matter of seconds to minutes. If it is taking a long time for the $CO_2$ signal to equilibrate, that suggests that $CO_2$ may be absorbing/desorbing onto the walls in your plumbing.

The tank is located downstairs indoors about 5 meters or so from the LGR, so there is some time required for the tubing as well as the LGR cavity to flush. Looking at the raw data, it takes somewhere on the order of 90-120 seconds to fully equilibrate. We were initially unsure how long the LGR needed to equilibrate, and wanted an idea of the variability/stability at the fixed concentration, which is why we initially ran the calibration gas for 60 minutes. It was switched later to conserve the breathing air tank.

What is the purpose of the Dasibi calibrator? Did you have to dilute the tank air to get ambient values?

The Dasibi calibrator is purely used as the scheduler for the tank used in the LGR stability evaluation. It contains a clock that turned the calibration gas on/off at the specified intervals. No dilution was performed, as the tank of breathing air provided a concentration within the normal range observed.

Figure 3: If you take the linear trend out, are the remaining variations related to a physical parameter such as temperature (ambient or cell)?

[Figure]

Residual Means vs Cavity T/P

The mean residuals from the linear trend plotted against the mean cavity temperature and pressure during the calibration periods show some correlation with pressure ($R^2$ of 0.27) and virtually no correlation with temperature ($R^2$ of 0.06)

Your drift correction technique of fitting a line to each subsequent pair of calibration points will introduce discontinuities into the corrected dataset that do not represent the real-world. It would be better to fit a smoothed curve (captures short-term drift) or a single linear fit (captures the long term drift).

Thank you for the excellent suggestion, using the single linear fit actually improves the RMSE slightly across all the sensors. The analysis throughout the paper (figures, table, numbers) will be update to reflect the results with the linear correction of the LGR drift.

When describing the differences between the K30 and LGR, you say that the LGR cavity temperature and pressure are relatively well controlled. Please give some numbers to give us a sense of how well controlled they are.

Over the entire evaluation period, the standard deviation of the 2-second data is 0.44 torr for cavity pressure and 0.06 °C for cavity temperature. This has been added to the manuscript.

You average the datasets into 1-minute bins based partially on the Allan variance results for the K30. Did you also do an Allan variance for the LGR?

Doing an Allan variance on the 0.5Hz LGR data using the breathing tank reveals that the noise is also Gaussian and that the optimum averaging interval is ~100 seconds, so 1 minute is also appropriate for the LGR data.

K30 Sensor Performance and Evaluation

Sect 2.1 – Did you compute an Allan variance for more than one sensor? Do they all perform similarly?

Yes, they all perform similarly. This has been added to the paper.

Figure 4 – $CO_2$ traces show periods of higher noise on some sensors (e.g. K30- 3 during the second half of the time period shown). In particular, I am wondering about the smattering of points that appear as outliers. In these cases, are the sensors still meeting the manufacturer's specification of +/- 30 ppm? Is there evidence that a sensor's precision can diminish over time?

Yes, there are some outliers of the two poor performing sensors that are outside of the ±30 ppm range from the original dataset, but after accounting for the zero offset, only K30-3 has any values outside of this range. This particular sensor has much more noise compared to the others, but for 1-sigma, it is within the specifications.

It's too difficult to say from this dataset if there is any evidence of precision diminishing over time. There is definitely a possibility as dust that could collect on the internal mirrors could change the absorption/path length of the IR light, and the light source could also potentially change with use. This would be something we hope to investigate in future work by performing a long-term (6-12 month) evaluation.

At the beginning of section 5, you state that $CO_2$ measurement differences are correlated with environmental variables, but you have not demonstrated the correlation. Can you show some scatter plots?

The correlations are not perfect by any means, but shown below are scatter plots for K30-1 for before the four stages of the regression:

For each plot the x-axis is the original K30 data – LGR. Top left: the difference versus the LGR

values, top right: the difference versus atmospheric pressure, bottom left: difference versus temperature, and bottom right: difference versus water vapor. Temperature is the least correlated (the K30 includes a crude first order temperature correction) and is the variable that has little effect on the RMSE (at least in this range of observed values), and pressure is the highest.  Note the density of points are not well resolved in the figures and will skew the fit lines, this is shown below with the second plot (larger figure).

[Figure]

[Figure]

Before doing an empirical fit to the environmental parameters, it would seem sensible to account for the dilution of the CO2 mixing ratio in humid air. See section 2 of Shusterman et al. 2016 for an example.

The multivariate regression takes into account the water vapor mixing ratio, as well as temperature and atmospheric pressure, so this should be accounted for when regressing against the $CO_2\_dry$ output from the LGR. The correction described in Sect. 2 of Shusterman et al. 2016 is essentially a simplified version of the multivariate regression where they correct for varying T,P,q.

Table 1 – Are all of the regression coefficients significant? Which parameter leads to the biggest improvement and which leads to the smallest improvement?

The most significant correction comes from the simple regression against the LGR reported CO2 values. Otherwise, because the sensor uses the absorption of infrared light, from the ideal gas law, it relates the concentration relative to a reference pressure, so atmospheric pressure has the largest correction. If you change the order, the final result is still the same, but since T/P/q are all correlated from weather and diurnal variations, the first one can often have the most significant impact.

Section 6.1 – Do you find that the K30 sensors that were closer to the LGR inlet have shorter lag times relative to the LGR response? You should try computing cross- correlation functions for each K30 against the LGR to improve the time-matching of the different time series.

Unfortunately, we do not have an exact record of the location or distance from each K30 to the LGR inlet. The cross-correlation functions would be difficult on this dataset because the lag can vary depending on things like time of day and weather, thus the lag correction may change throughout the time series. In a real-world setting, there could be occasional dramatic shifts in concentrations from plumes, boundary layer dynamics, or other reasons, so this demonstrates that there is a need to wait several minutes until the signal equilibrates somewhat.

Significant figures - Most of the performance metrics stated for the K30 sensors in units of ppm CO2 are given with two decimal places, yet you state the K30 measurement resolution is only 1 ppm.

Yes, the output is 1 ppm resolution, but for 2-second data, when using the 1-minute averages, the effective resolution is higher. We have now changed these units to use only one decimal place, which is more consistent with the precision of the 1-minute averages but still shows the impact the regressions have on the RMSE.

Figures 5,6,7,8,10 are all shown for K30 #1, which, from Figure 4, appears to be one of the best performing sensors. I would be curious to see a residual plot for sensor #3, 5 or 6.

Here is Figure 8, but rather for K30 #5 than K30 #1.

[Figure]

You state that one goal of this work is to understand how correction coefficients can be derived quickly. Wouldn't it be more efficient to design a controlled experiment where controlling variables are deliberately varied across the full range of operating conditions?

We were looking for a way to derive the coefficients for a large group of sensors with minimum human labor. By artificially controlling air temperature, pressure, and moisture content, the cost of the evaluation both in time and money would increase, negating some of the benefits of the price of these sensors.

Section 7 – In future work, you aspire to characterize the sensors' maximum performance in a controlled environment. Yet, if the big-picture goal is to use these sensors is to generate science quality ambient air measurements, I believe a more worthy goal would be to characterize their minimum performance in an uncontrolled environment.

This is another area we hope to pursue with future work, but would need to devise a way that would be uncontrolled but also meaningful enough for publishable data. Ideally we would like some installed outdoors next to an inlet for a gas analyzer as the reference, but would need to ensure that the sensors are in an enclosure that provides adequate ventilation but also protection from weather.

Technical comments

Title – I don't think "enhancement" is the right word. How about something like "Evaluation and correction of CO2 measurement in ambient air from a low-cost sensor"

Thanks for the suggestion, we changed it to "Evaluation and environmental correction of ambient $CO_2$ measurements from a low-cost NDIR sensor"

Abstract – The quantities reported have different numbers of significant figures. These should be uniform and reflect the precision of the measurement.

As mentioned above, we now use one decimal place for the RMSE of the K30 sensors.

Pg 1, Ln 25 – "dry air" is used twice in this sentence.

Fixed.

Pg 1, Ln 28 – The WMO compatibility goal is a goal, but is not always achieved, and certainly not for historical measurements.

We agree with the reviewer. We have changed this to state that this is the WMO compatibility goal.

Pg 1, Ln 29 – Suggest: '. . .to collect samples, which are subsequently transported'

Thanks, changed it to reflect this suggestion.

Pg 1, Ln 30 – You mention two expensive types of measurements – flasks and Picarros, but you do not mention moderately-priced analyzers from LiCor and Los Gatos, which are used at many research-grade monitoring sites.

This is true, but the added costs for calibration and maintenance can make a LiCor (and Los Gatos as you've seen from our analysis) comparable in total cost to a Picarro. The models are not explicitly stated and this is merely to show that either flasks or continuous observations are prohibitively expensive to do at high spatial resolution. We have added a phrase to indicate that this cost includes labor and calibration costs.

Pg 2, Ln 1 – Be consistent about whether you spell out carbon dioxide or use the abbreviation.

This has been corrected throughout the manuscript. Thanks.

Pg 2, Ln 2 – Suggest paragraph break at "Recent research"

Done.

Pg 2, Ln 8 – Is 8-12 sites typical? There are ~5 sites in Boston, SLC, and Paris.

For the LA Megacities project there are 14 sites, 11 current sites in Indianapolis, and a planned 14 sites for DC/Baltimore, 8-12 was used as a rough average of these 6 cities. The text has been changed to 3-12 to reflect the inclusion of the smaller networks listed. Additional references have been added here to show a sample of these networks both in size and geographic location.

Pg 2, Ln 9 – You say that more dense observations, even with larger uncertainties, yield better inversion constraints, but this is all relative and depends on the inversion setup/goals. See Turner et al., 2016, ACP for an exploration of the tradeoffs.

The text has been updated to state that this depends on the methodology used, and a citation to Turner et al., 2016 has been added.

Pg 2, Ln 14 – Suggest deleting "however"

Done.

Pg 2, Ln 16 – Suggest changing the phrasing to: "Recent evaluations and implementations of new low-cost sensors demonstrate their promise for ambient air monitoring."

Changed to "Evaluation and implementation of some of these new low-cost sensors demonstrate their promise for ambient air monitoring."

Pg 2, Ln 26 – Can you give some numbers to scope what you mean by "reasonably accurate"?

Based on the cited texts, ±3-5ppm, has been added to the manuscript.

Pg 3, Ln 7 – Suggest: "The K30 sensor module from SenseAir (Sweden) is the low-cost NDIR CO2 sensor that was tested for this study".

Changed to "The K30 sensor module (K30) from SenseAir (Sweden), is the low-cost NDIR $CO_2$ observing instrument used in this study[1]."

Pg 3, Ln 10 – Suggest deleting "given as"

Thanks. Changed.

Pg 3, Ln 13 – Suggest: "The K30 was chosen not only because it has the highest manufacturer-specified accuracy, but also because initial testing showed reliability and consistency with higher-quality observations."

Changed to close to your suggestion: "The K30 was chosen not only because of it has the highest manufacturer-specified accuracy, but also because initial testing showed reliability and

consistency when compared to higher-quality observations."

Pg 3, Ln 17 – You should give the units (relative humidity) for the 3% and 0.008% quantities.

Thanks for the suggestion. We have revised the sentence as "…has an average absolute accuracy of ±1 ℃, ±3 %, and ±1 hPa, and an output resolution of 0.1 ℃, 0.008 % and 0.01 hPa for temperature, relative humidity, and pressure, respectively"

Pg 3, Ln 24 – "less than one percent" "< 1%"

Fixed.

Pg 4, Ln 5 – Another difference between the two analyzers could be their sensitivity to the isotopes of CO2.

This could be true, but without knowing for sure, we prefer to not add this point to the paper. The LGR is only sensitive to $^{12}C$, but the standards used to calibrate the LGR account for all isotopes of $CO_2$. Additionally, the component of $^{13}C$ is around 1% relative to $^{12}C$, and thus the difference would be small. This is now briefly addressed in the text.

Pg 4, Ln 29 – Can you briefly describe what you mean by "various complications"?

The Raspberry Pi runs a full Linux OS, so because of the complexity of the OS, sometimes there is a delay in when certain tasks execute, which may compound into some sensors collecting (at times) observations out of phase from others. The LGR 0.5Hz data starts whenever the system initializes. Thus perfect synchronization is difficult, but all have recorded time stamps and can be averaged / regularized for comparison. A brief explanation of this has been added to Section 2.

Pg 5, Ln 1 – My understanding is that you merged datasets by their timestamps. Did you have to do something to keep the clocks synchronized?

All of the Raspberry Pi data loggers use an internet server to synchronize their time, and the LGR uses an internal clock with battery that was set to the same time as the Pis at the beginning of the experiment. This has been added to Section 2.

Pg 5, Ln 13 – "longer averaging times do not reduce the noise"

Fixed. Thank you for noticing this.

Pg 7, Ln 21 – Suggest deleting "However".

Done.

Pg 7, Ln 22 – Is the statement about each K30 meeting the manufacturer's uncertainty specification in regards to the raw (2-second) data or 1-minute averages? Please clarify in the text.

The manufacturer specifies the range for the raw data but our analysis is for the 1-minute averages. Text is updated to reflect this, both in section 2 that the datasheet is for 2-sec and in section 4 that our analysis is for 1-min.

Pg 10, Ln 21 – suggest: ". . . and 1.48 ppm, for 1-minute, 10-minute, and hourly averages, respectively.

Text is changed to this suggestion.

Pg 10, Ln 26 – suggest: "One goal of this work is to develop a methodology to evaluate individual sensors quickly. . ."

Changed to "One goal of this work is to develop a methodology to evaluate individual sensors quickly so that they can be used in scientific applications."

Pg 11, Ln 32 – "less than five parts per million" "< 5 ppm"

Fixed here as well.

Figure 1 – A ballpoint pen is included in the picture for size reference. A ruler instead of a pen would be more useful.

We liked this idea, and Figure 1 now includes a ruler instead of a ballpoint pen.

Figure 2 – What was the $CO_2$ concentration of the tank used?

This is the breathing air tank used for the LGR drift, so estimated to be 463.7 ppm after calibrating the LGR with NIST standards, as noted in the text at the end of Section 3.

Figure 4 – State the time interval of the data shown. I can't tell if this is raw 2-second data or 1-minute averages.

1-minute averages, all figure captions have been updated to clarify this.

Figure 8 – I don't understand the difference between the red and blue points.

The blue data points are used in the regression, and the red is the complete dataset. This is done for consistency with internal plots that show the time series for regression periods of varying

length. Captions for Figs. 8 and 10 have been updated to clarify this.

Figure 9 – Can you put error bars on each point for the y-variable?

We decided to instead show a box plot as well as all six sensors' values, as this gives a better picture of the variability than just error bars. Please see updated Figure 9.

Response specific citations:

Yang, M., Prytherch, J., Kozlova, E., Yelland, M. J., Parenkat Mony, D., and Bell, T. G.: Comparison of two closed-path cavity-based spectrometers for measuring air–water $CO_2$ and $CH_4$ fluxes by eddy covariance, Atmos. Meas. Tech., 9, 5509-5522, doi:10.5194/amt-9-5509-2016, 2016.

---

## Referee Report (RR1)

You currently use either of two acronyms for the Los Gatos. I suggest sticking with just one, preferably "LGR", which is more intuitive.

Pg 2, Line 1 – The addition to this sentence doesn't make sense the way it is currently phrased since labor and calibration cases are not part of the analyzer cost. Suggest: "…high-precision analyzers can cost upwards of $100,000 per site, plus any additional costs for labor, calibration gases and installation of equipment such as inlet lines."

Pg 2, Line 13 – Suggest for clarity: "Observing system simulation experiments (OSSE) have found that a higher spatial density of observations in these urban regions could better constrain emission estimates, even if the absolute uncertainty of the observations is higher (Turner et 15 al., 2016; Wu et al., 2016; Lopez-Coto et al., under review), but a trade-off between total network cost and inversion constraint must be balanced, and this result depends on the assumptions of the OSSE."

Pg 2, Line 34 – "precision"; do you mean accuracy?

Pg 3, Line 14 – Suggest: "…including the COZIR ambient sensor and Telaire T6615, which have manufacturer specified accuracies of ±50 ppm ±3 % and ±75 ppm respectively."

Pg 7, Line 5 – Suggest for clarity: "using a reference tank of breathing air connected to a Dasibi Model 5008 calibrator, which was used to schedule the input of calibration gas"

Pg 7, Line 4 – Suggest: "…for a period of one hour, initially, and later, ten minutes, to conserve…"

Pg 9, Line 4 – was → were

Pg 11, Line 23 – Suggest: "…but the coefficients are calculated using only data from the first 15 days."

Page 12, Lines 3, 5, 7 – uniform → generalized

Figure 8 – The points used (blue) in the top panel represent the vast majority of points. Showing the extra red points at the end of the timeseries seems to add unnecessary complexity to the plot. Why not just cutoff the plot at 25 days or else use all the points in the regression?

Figure 9 – Could add a legend with K30 number so we can relate it back to Fig 4, Table 1.

Section 6.3 – To compute a generalized equation, I would have grouped the data from all sensors and fit a single regression. Does this yield the same result as averaging the coefficients from the individual fits as you did?

---

## Author Response (AR2)

**Second response to Anonymous Referee #1**

Responses in blue

You currently use either of two acronyms for the Los Gatos. I suggest sticking with just one, preferably "LGR", which is more intuitive.

Done. Now LGR is used exclusively to refer to the Los Gatos analyzer.

Pg 2, Line 1 – The addition to this sentence doesn't make sense the way it is currently phrased since labor and calibration cases are not part of the analyzer cost. Suggest: "...high-precision analyzers can cost upwards of $100,000 per site, plus any additional costs for labor, calibration gases and installation of equipment such as inlet lines."

The sentence has been changed to: "Continuous in-situ $CO_2$ analyzers located at towers do not suffer from these regular costs, but these high-precision analyzers can cost upwards of $100,000 per site, plus any additional costs for calibration gases and installation of equipment and inlet lines."

Pg 2, Line 13 – Suggest for clarity: "Observing system simulation experiments (OSSE) have found that a higher spatial density of observations in these urban regions could better constrain emission estimates, even if the absolute uncertainty of the observations is higher (Turner et al., 2016; Wu et al., 2016; Lopez-Coto et al., under review), but a trade-off between total network cost and inversion constraint must be balanced, and this result depends on the assumptions of the OSSE."

We agree that it is a good idea to clarify that OSSEs show this, and thus the sentence has been changed to "Observing system simulation experiments have found that, depending on the methodology used, a higher spatial density of observations in these urban regions has been shown to better constrain the inversion estimates, even if the absolute uncertainty of the observations is higher (Turner et al., 2016; Wu et al., 2016; Lopez-Coto et al., in press), but a trade-off between total network cost and inversion constraint must be balanced."

Pg 2, Line 34 – "precision"; do you mean accuracy?

Yes, changed to accuracy.

Pg 3, Line 14 – Suggest: "...including the COZIR ambient sensor and Telaire T6615, which have manufacturer specified accuracies of ±50 ppm ±3 % and ±75 ppm respectively."

Thank you for the suggestion, the sentence has been changed accordingly.

Pg 7, Line 5 – Suggest for clarity: "using a reference tank of breathing air connected to a Dasibi Model 5008 calibrator, which was used to schedule the input of calibration gas"

Thanks again, it is a good idea to clarify, and has been added to the text.

Pg 7, Line 4 – Suggest: "...for a period of one hour, initially, and later, ten minutes, to conserve..."

Has been changed to suggestion.

Pg 9, Line 4 – was -> were

Thank you for finding this, has been changed.

Pg 11, Line 23 – Suggest: "...but the coefficients are calculated using only data from the first 15

days."
Great suggestion, and the text has been changed to this.

Page 12, Lines 3, 5, 7 – uniform -> generalized

Changed. Thank you.

Figure 8 – The points used (blue) in the top panel represent the vast majority of points. Showing the extra red points at the end of the timeseries seems to add unnecessary complexity to the plot. Why not just cutoff the plot at 25 days or else use all the points in the regression?

This is a good point. The blue points were used to create a batch of plots for varying regression length, but for the publication, it makes more sense to just use the entire period and remove the blue points. Figure 8 has been updated to reflect this.

Figure 9 – Could add a legend with K30 number so we can relate it back to Fig 4, Table 1.

Done.

Section 6.3 – To compute a generalized equation, I would have grouped the data from all sensors and fit a single regression. Does this yield the same result as averaging the coefficients from the individual fits as you did?

Computing a regression for all sensors at once gives slightly different coefficients than the average of the individual regressions, but it yields a similar result. The RMSEs are still quite poor compared to when using individual regression coefficients, but their spread is smaller than before (5.2-19.8 ppm rather than 3.1-23.9 ppm). A sentence has been added to the manuscript to include this info.

[revised manuscript text omitted]